

# Validating Precision Estimates in Horizontal Wind Measurements from a Doppler Lidar

Rob K. Newsom[1], W. Alan. Brewer[2], James M. Wilczak[2], Daniel E. Wolfe[2,3], Steven P. Oncley[4], Julie K. Lundquist[5,6]

[1]Pacific Northwest National Laboratory, Richland, WA, 99352, USA
[2]National Oceanic and Atmospheric Administration, Earth System Research Laboratory, Boulder, CO, 80305, USA
[3]Cooperative Institute for Research in Environmental Sciences, Boulder, CO, 80305, USA
[4]National Center for Atmospheric Research, Boulder, CO, 80307, USA
[5]Department of Atmospheric and Oceanic Sciences, University of Colorado, Boulder, CO, 80309, USA
[6]National Renewable Energy Laboratory, Golden, CO, 80401, USA

*Correspondence to*: Rob K. Newsom (rob.newsom@pnnl.gov)

**Abstract.** Results from a recent field campaign are used to assess the accuracy of wind speed and direction precision estimates produced by a Doppler lidar wind retrieval algorithm. The algorithm, which is based on the traditional velocity-azimuth-display (VAD) technique, estimates the wind speed and direction measurement precision using standard error propagation techniques. For this study, the lidar was configured to execute an 8-beam plan-position-indicator (PPI) scan once every 12 minutes during the 6 week deployment period. Several wind retrieval trials were conducted using different schemes for estimating the uncertainty in the radial velocity measurements. The resulting wind speed and direction precision estimates were compared to differences in wind speed and direction between the VAD algorithm and sonic anemometer measurements taken on a nearby 300-m tower.

All trials produced qualitatively similar wind fields with negligible bias, but substantially different wind speed and direction precision fields. The most accurate wind speed and direction precisions were obtained when the radial velocity uncertainty was determined by direct calculation of radial velocity standard deviation along each pointing direction and range gate of the PPI scan. By contrast, setting the radial velocity uncertainty to the radial velocity precision (thereby ignoring turbulence effects) resulted in wind speed and direction precisions that were biased far too low and poor indicators of data quality.

## 1 Introduction

Coherent Doppler Lidars (CDL) are used in applications ranging from basic atmospheric boundary-layer research, to model data assimilation (Riishøjgaard, et al., 2004; Chai, et al. 2004; Newsom and Banta 2004a; Newsom and Banta 2004b; Newsom et al., 2005; Weissmann and Cardinali, 2007; Pu et al., 2010), to wind resource assessment (Pena, et al., 2009; Lang et al., 2011; Koch et al., 2012; Pichugina et al., 2012; Hsuan et al., 2014; Newsom, et al. 2015; Newman, et al. 2016;





Choukulkar A., et al., 2016). Within the wind energy industry, these instruments are viewed as cost effective alternatives to instrumented towers for wind resource assessment provided the measurement uncertainties are within acceptable limits.

Quantifying the uncertainty in CDL-derived wind speed and direction measurements is perhaps best achieved by comparison with collocated in situ reference measurements. For wind resource assessment, the International Energy Agency's recommended best practices (Clifton et al 2013) state that CDLs should undergo periodic verification using anemometers that have been calibrated against a traceable standard. This enables determination of uncertainty due to both the random error (i.e. precision) and systematic error (i.e. bias). Although these verification studies are valuable, one problem is that they are invariably carried out at locations different from the resource assessment site. Thus, the uncertainties determined in this manner may or may not be applicable to the measurements obtained at the resource assessment site. It is, however possible to obtain "on-site" estimates of the wind speed and wind direction precision using standard error propagation techniques in the CDL wind retrieval algorithm if the variability in the radial velocity measurements can be properly characterized. These precision estimates could then be used as measures of data quality.

A number of studies have been performed to characterize uncertainties in CDL wind retrievals (Gottschall et al. 2012; Lane et al. 2013; Davies et al. 2003). Lundquist et al. (2015) examined errors caused by inhomogeneous flow using a lidar simulator approach in which high-resolution LES output was used as the reference field. Wang et al. (2016) used field measurements to investigate uncertainties in winds retrieved from sector scans and found the radial velocity variance to be a robust measure of the uncertainty in the retrieved winds due to its relationship with turbulence.

In this study, we evaluate the accuracy of CDL-derived wind speed and wind direction precision estimates obtained by propagating radial velocity uncertainties through a velocity-azimuth-display (VAD) algorithm. The VAD algorithm uses data from a plan-position-indicator (PPI) scan to generate estimates of wind speed and direction as a function of height. PPI scans are performed by scanning the lidar beam in azimuth while maintaining a constant elevation angle. A key assumption in the algorithm is that the flow is horizontally homogeneous. For perfectly horizontally homogeneous flow in the absence of measurement error, the radial velocity versus azimuth curve forms a perfect sinusoid. In reality, deviations from the perfect sinusoid occur due to spatial and temporal fluctuations in the velocity field, and instrumental errors. In the context of the VAD algorithm, any departure from the perfect sinusoid may be regarded as error. This study evaluates the accuracy of wind speed and direction precision estimates using three different methods of estimating the radial velocity uncertainty. This is accomplished using data that were acquired during the eXperimental Planetary boundary-layer Instrument Assessment (XPIA) field campaign (Lundquist et al. 2016) at the Boulder Atmospheric Observatory (BAO).

During XPIA, several CDL systems were deployed on or near the BAO site for the purpose of assessing the accuracy of single- and multiple-Doppler retrievals of 3D wind fields. The 300-m BAO tower was instrumented with pairs of sonic anemometers on opposing booms at six levels, thus providing accurate reference measurements against which the CDL wind



retrievals could be compared. In this study, we focus specifically on the wind speed and direction retrievals from a single CDL system.

The CDL used in this study was on loan from the US Department of Energy's Atmospheric Radiation Measurement (ARM) program (Mather and Voyles 2014), and represents one of nine CDL systems that ARM currently operates at various sites

around the world in support of climate research. The CDLs at the ARM sites spend the bulk of their time staring vertically acquiring high-temporal-resolution (~1 Hz) measurements of vertical velocity. The vertical stares are momentarily interrupted once every 10 to 15 minutes to perform an 8-beam step-stair plan-position-indicator (PPI) scan, from which profiles of horizontal wind speed and direction are derived. This simple scanning strategy was designed to provide well-resolved measurements of the horizontal winds while minimizing gaps in the vertical velocity time series, which is essential

for computing turbulent power spectra. During XPIA, the CDL used in this study was operated in a similar manner so that the results could be readily transferred to ARM operations.

We also investigate the effect of scan geometry and dimensionality (2D vs 3D) of the retrieved winds and their uncertainties. Trials were conducted in which the VAD algorithm was configured to perform retrievals using different 4-beam subsets of the available 8 beams from each PPI scan. If the same (or better) performance can be achieved using few beams then it

would be possible to reduce the time spent doing PPI scans, thereby reducing the gap in the vertical velocity time series. Trials were also conducted to evaluate the impact of neglecting the vertical velocity in the retrieval algorithm.

This paper is organized as follows. Section 2 describes the experimental setup, including a brief description of the instrumentation on the 300-m BAO tower as well as the configuration and scan strategy used by the CDL. The lidar wind retrieval algorithm and error propagation methods are presented in Section 3. Section 4 presents the results of the wind

retrieval trials using the sonic anemometers on the 300-m BAO tower as reference measurements. This discussion includes three trials that were performed using different schemes for estimating the radial velocity uncertainty, as well as five additional trials that were performed to examine the impact of scan geometry and dimensionality (2D vs 3D) on the retrieved winds and their uncertainties.

## 2 Experimental Setup and Instrumentation

Prior to decommissioning in 2016, the BAO was a research facility maintained by NOAA's Earth System Research Laboratory (Kaimal and Gaynor, 1983). The facility was located on Colorado's high plains approximately 25 km east of Boulder, Colorado (40.05°N, 105.00°W). The center piece of the BAO site was a 300-m meteorological tower. Figure 1 shows the set up during XPIA and the location of the CDL relative to the BAO tower, 139 m south of the tower.



## 2.1 Doppler Lidar

The CDL used in this study is a commercial grade system manufactured by Halo Photonics. This particular system was first acquired by ARM in 2010, and was subsequently deployed to ARM's Tropical Western Pacific site in Darwin Australia, before that site was decommissioned at the beginning of 2015. The system is currently one of five CDLs operating at ARM's

Southern Great Plains site in north-central Oklahoma.

The CDL provides range-resolved measurements of attenuated aerosol backscatter, signal-to-noise ratio (SNR), and radial velocity. The system employs an eye-safe laser that transmits at a wavelength of 1.548μm, with ~150ns (22.5m) <100μJ pulses at a rate of 15kHz. The primary scattering mechanism is atmospheric aerosol. As a result, valid measurements are usually restricted to the atmospheric boundary layer where aerosol concentrations are high enough to ensure good signal-to-

noise ratio. The instrument incorporates a full upper hemispheric scanner that can be configured to operate in either step-stare or continuous scan mode. In step-stare mode, the CDL acquires data by dwelling with the beam in a fixed pointing direction, and scans are performed by incrementally moving from one pointing direction to the next. In continuous scan mode, the CDL acquires data continuously as the scanner moves between predefined starting and ending positions. The internal processor can be configured to operate using a wide range of pulse accumulation times and range gate sizes. During

XPIA, the system was operated using a range gate size of 30m, and 200 range gates, resulting in a maximum measurement range of about 6 km. Typically, the maximum range for usable measurements varied between 1 to 3 km, depending on atmospheric conditions. Further details about the ARM Doppler lidars can be found in Pearson et al. (2009), and Newsom (2012).

From 6 March to 16 April 2015 the ARM CDL was deployed within a small cluster of profiling CDLs approximately 140 m

south of the BAO tower, as shown in Figure 1. During this time, the instrument was operated using a fixed scan schedule consisting of PPI scans once every 12 minutes, 10-minute tower stares once per hour, and target sector scans once per day. The remainder (and majority) of the time was spent acquiring 1 Hz vertical velocity profiles.

PPI scans were performed using the step-stare scan mode at an elevation angle of 60° and eight evenly-spaced azimuth angles around the compass, as illustrated in Figure 2. The pulse integration time for each profile was set to 2 s (30000 laser

pulses), and the time required to execute one complete PPI scan was about 40 seconds. We note that this is the same configuration that is currently used by most of the CDLs at the ARM sites.

The pointing accuracy of the lidar beams is crucial for determination of the wind direction. Although the angular precision of the scanner is about ±0.05°, the pointing accuracy requires careful calibration. To ensure proper orientation relative to the vertical coordinate, the 'levelness' of the instrument was routinely monitored throughout the deployment period, and target

sector scans were performed in order to accurately determine the lidar's azimuthal pointing direction relative to true north. The target scan consisted of a high-angular-resolution narrow-sector continuous mode PPI scan in the general direction of a



target of opportunity. The observed location of the hard target return in the scan data, together with the known GPS coordinates of the lidar and the target enabled determination of the lidar's orientation with respect to true north. For this experiment the target that was used was a tall stadium light post located next to the football field at Erie High School, at a distance of about 800m west of the lidar location.

**2.2 BAO Tower Sonic Anemometers**

During XPIA the BAO tower was instrumented at six levels (50, 100, 150, 200, 250, and 300m) with fast response (20Hz) 3-D sonic anemometers (Campbell CSAT3). Each level had two sonic anemometers, one mounted on a southeast boom (at a heading angle of $154^{o}$) and one mounted on a northwest boom (at a heading angle of $334^{o}$). This improved the odds of obtaining measurements that were unaffected by the tower wake. Data from the sonic anemometers were tilt-corrected

(Wilczak et al. 2001), and rotated into geographical coordinates, with positive $u$ toward the east, positive $v$ toward the north, and $w$ pointing in the corrected vertical direction.

Figure 3 summarizes mean wind statistics as determined from 10-min average sonic anemometer data during the ARM CDL deployment period from 6 March through 16 April, 2015. The wind rose shown in Fig 3a indicates that the strongest winds tended to blow from the northeast and the west. Although there was no strongly preferred wind direction, there was a slightly

higher occurrence of northeasterly flow during the deployment period. Winds were generally fairly light with the bulk of the wind speeds occurring between 3 and 6 ms$^{-1}$. Fig 3b indicates that the median wind speeds tended to be at or slightly below 4 ms$^{-1}$. Inspection of the data shows that wind speeds exceeding 15ms$^{-1}$ occurred for brief periods on 16-17 March, 24-25 March, 12 April and 15-16 April. Overall, however, the winds rarely exceeded 10 ms$^{-1}$ during the deployment period.

**3 Lidar Wind Retrieval and Precision Estimation**

CDL estimates of horizontal winds are computed from PPI scan data using a velocity-azimuth-display (VAD) algorithm based on the classic technique described by Browning and Wexler (1968). Assuming the flow to be horizontally uniform and steady at a given range gate or height above the ground, the wind velocity components are retrieved by fitting a sinusoid to the radial velocity data; the amplitude, phase and offset of the sinusoid determine the wind speed, wind direction and vertical velocity, respectively. This is equivalent to minimizing

$$\psi^2 = \sum_{i=1}^{N} \frac{\left(\mathbf{u}_o \tilde{\mathbf{r}}_i^T - u_{ri}\right)^2}{\sigma_{ri}^2}$$

(1)



with respect to the components of the mean velocity vector, $\mathbf{u}_o = (u_o, v_o, w_o)$. In equation (1), $u_{ri}$ is a radial velocity

measurement, $\sigma_{ri}$ is the measurement uncertainty due to random errors, and $\tilde{\mathbf{r}}_i$ is a unit vector from the lidar to the

measurement point, i.e. the beam pointing direction, and is given by

$$\tilde{\mathbf{r}}_i = \left( \sin\phi_i \cos\theta, \cos\phi_i \cos\theta, \sin\theta \right) \qquad (2)$$

5    where $\phi_i$ is the azimuth angle as measured clockwise from true north, $\theta$ is the (constant) elevation angle as measured from

the horizontal plane, and $\tilde{\mathbf{r}}_i^T$ is the transpose of $\tilde{\mathbf{r}}_i$. The summation in equation (1) is performed over all the pointing

directions in the PPI scan. Minimizing equation (1) with respect to the components of $\mathbf{u}_o$ results in a system of three

equations and three unknowns, the solution of which can be expressed as

$$\mathbf{u}_o = \mathbf{C}\mathbf{b}, \qquad (3)$$

10   where

$$\mathbf{C} = \left( \sum_{i=1}^{N} \frac{\tilde{\mathbf{r}}_i^T \tilde{\mathbf{r}}_i}{\sigma_{ri}^2} \right)^{-1} \qquad (4)$$

is the covariance matrix, and

$$\mathbf{b} = \sum_{i=1}^{N} \frac{u_{ri}}{\sigma_{ri}^2} \tilde{\mathbf{r}}_i^T . \qquad (5)$$

Equation 3 determines the wind velocity components at a fixed range gate. The height of the range gate above ground level

is given by $z = r\sin\theta$, where $r$ is the distance from the lidar to the range gate center. Wind profiles are then constructed

by applying equation (3) to all range gates.

When the individual radial velocity uncertainties, $\sigma_{ri}$, are known the precision of the retrieved velocity components can be

20   obtained from the diagonal elements of the weighted covariance matrix (Press et al. 1988), i.e.



$$\sigma_u = \sqrt{C_{11}} \quad \text{and} \quad \sigma_v = \sqrt{C_{22}} \ . \tag{6}$$

When the radial velocity uncertainties are not known, the precisions in $u$ and $v$ can be estimated by setting $\sigma_{ri} = 1$ in equation (1). The precisions of the retrieved velocity components are then determined in the following manner (Press et al. 1988):

$$\sigma_u = \psi \sqrt{\frac{C_{11}}{N - N_f}} \quad \text{and} \quad \sigma_v = \psi \sqrt{\frac{C_{22}}{N - N_f}} \ , \tag{7}$$

where $N_f$ is the number of retrieved parameters, i.e. $N_f = 3$ for 3D retrievals ($u_o$, $v_o$ and $w_o$) or $N_f = 2$ for 2D retrievals ($u_o$ and $v_o$ only).

Given the precisions for $u_o$ and $v_o$, the estimated uncertainty in the retrieved wind speed is given by

$$\sigma_M^{est} = \left( (u_o \sigma_u)^2 + (v_o \sigma_v)^2 \right)^{1/2} / M \tag{8}$$

10 where $M = \sqrt{u_o^2 + v_o^2}$ is the CDL wind speed, and the estimated wind direction precision is given by

$$\sigma_\alpha^{est} = \left( (u_o \sigma_v)^2 + (v_o \sigma_u)^2 \right)^{1/2} / M^2 . \tag{9}$$

The key assumption underlying the retrieval method is that the mean winds are horizontally homogeneous over the time it takes to perform a PPI scan. The actual measured radial velocity contains contributions from turbulent velocity fluctuations, $\mathbf{u}'$, and instrumental noise, $n$, such that

$$u_r = \left( \mathbf{u}_o + \mathbf{u}' \right) \tilde{\mathbf{r}}^T + n \tag{10}$$

For any given pointing direction at a fixed range from the lidar we assume that the ensemble average of the turbulent and noise terms are zero, and that the noise and turbulence are completely uncorrelated. In that case, the variance of the measured radial velocity can be written as

$$\sigma_r^2 = \overline{u_r'^2} + \sigma_n^2 \tag{11}$$



where $u'_r = \mathbf{u}'\tilde{\mathbf{r}}^T$ is the contribution from turbulence, and $\sigma_n$ is the standard deviation of the instrumental noise, which we refer to as the radial velocity measurement precision.

The radial velocity measurement precision is defined as the uncertainty (due to random errors) in the absence of any atmospheric variability. This uncertainty, which increases with decreasing SNR, is caused by random fluctuations in the

Doppler shift estimates due primarily to frequency drift in the local oscillator (Frehlich et al. 2004). The performance of the CDL system can be partly characterized by parameterizing the measurement precision in terms of the SNR. This results in an empirical curve that is independent of range and atmospheric conditions.

Figure 4a shows the radial velocity measurement precision as a function of the SNR for the ARM CDL. Precision estimates were obtained through analysis of the vertical staring data using the autocovariance technique described by Lenschow et al.

(2000) and Pearson et al (2009). For this study, the precision and mean SNR were computed from 30-min time series of radial velocity. This analysis was performed using all contiguous 30-minute time periods and range gates in the vertical staring data set collected during XPIA. This resulted in a large number of precision and mean SNR sample pairs. The result shown in Fig 4a was obtained by fitting a smooth curve through these sample pairs.

All wind retrievals in this study were processed by requiring that the SNR be greater than or equal to 0.008 for each beam in

a given PPI scan. This threshold level, which is indicated by the dotted and dashed vertical lines in Fig 4, corresponds to a radial velocity precision value of roughly $1\text{ms}^{-1}$. As a result, radial velocities with $\sigma_n$ greater than approximately $1\text{ms}^{-1}$ were rejected and thus not used to compute the winds. We note that the SNR over the height of the BAO tower was almost entirely above threshold during the deployment period, as illustrated by the median SNR profile in Fig 4b.

## 4 Results

### 4.1 Uncertainty Estimation Methods

Three trials were conducted to evaluate different methods for estimating the radial velocity uncertainty and the resultant uncertainty in the derived wind components. For all of these trials the wind retrieval algorithm was configured to retrieve all three components of the wind field and use all eight beams of the PPI scans. These trials are described in Table 1.

Trial 1 uses the simplest method for estimating the precision, and is currently the method used to process PPI scan data from

the all the existing ARM CDLs. We note that this approach is equivalent to assuming that the radial velocity uncertainty is independent of the azimuth, i.e. isotropic, and thus independent of the mean wind direction.





For Trial 2 radial velocity uncertainty for a given pointing direction and range gate is estimated by computing the variance of the radial velocity over three consecutive scans and two neighboring range gates. Thus, for the $i^{th}$ azimuth, $j^{th}$ range gate, and $q^{th}$ PPI scan the radial velocity uncertainty for Trial 2 is estimated from

$$\sigma_r^q(\phi_i, r_j) = \left[ \frac{1}{9} \sum_{l=q-1}^{q+1} \sum_{k=j-1}^{j+1} \left( u_r^l(\phi_i, r_k) - \overline{u}_r^q(\phi_i, r_j) \right)^2 \right]^{1/2}, \tag{12}$$

where

$$\overline{u}_r^q(\phi_i, r_j) = \frac{1}{9} \sum_{l=q-1}^{q+1} \sum_{k=j-1}^{j+1} u_r^l(\phi_i, r_k). \tag{13}$$

The idea behind this scheme was to approximate a scanning strategy in which several independent radial velocity samples are recorded for each pointing direction and range gate. This provides a means of estimating the uncertainty directly from the observations and avoids the assumption of isotropy inherent in Trial 1. We emphasize that the averaging scheme described
above is only used to estimate the radial velocity uncertainty. The radial velocity measurements used in Trial 2 are the same as those used in the other Trials.

For Trial 3 the radial velocity uncertainty, $\sigma_r$, is taken to be equal to the radial velocity measurement precision, $\sigma_n$, so that the effects of turbulence are completely ignored. Since the radial velocity precision is parameterized in terms of the SNR (see Fig 4a), the radial velocity uncertainty for Trial 3 is determined solely by the SNR.

It is important to note that all of the trials described above use the same radial velocity data. The only difference between Trials 1, 2 and 3 is in the treatment of the radial velocity uncertainty, $\sigma_r$.

Figure 5 shows representative lidar wind retrievals for 4 April 2015, with results for Trials 1, 2 and 3 shown in panels (a), (b) and (c), respectively. The wind speed and direction retrievals (left) for all three trials look qualitatively similar, but the precision estimates (right) are substantially different, particularly between Trial 3 and either Trial 1 or 2. The precision for
Trials 1 and 2 both show a strong diurnal dependence with larger precisions during the daytime period (sunrise and sunset are about 12 and 2 UTC, respectively). The uncertainties for Trial 2 are larger than Trial 1 and exhibit more smoothing as a result of the averaging described by equation (12). By contrast, the uncertainties for Trial 3 are much smaller than either Trials 1 or 2 and exhibit a completely different structure with no distinctive diurnal variation.





To enable comparison with the CDL-derived winds, 20Hz data from each sonic anemometer on the BAO tower were averaged in time and then interpolated to the heights of the lidar range gates closest to the sonic anemometer heights. The temporal averaging procedure used scalar averaging for the wind speed and vector averaging for the wind direction. The center times of the averaging intervals were made to coincide with the center times of the PPI scans, and the durations of the averaging intervals were set equal to twice the PPI scan durations. We note that this method results in an under-sampling of the sonic anemometer data, since the time between PPI scans is 12 min and the nominal scan duration is about 40 sec.

The temporally averaged sonic anemometer wind profiles were then interpolated to the heights of the lidar range gates closest to the sonic anemometer heights (these heights were 142.9, 194.9, 246.8 and 298.8 m). The interpolation was handled by interpolating the horizontal vector components, rather the wind speeds and direction. Also, only those anemometers on the upwind side of the tower were used in the interpolation as tower wake effects were observed to be quite significant (McCaffrey, et. al. 2016).

The interpolation of the sonic measurements to the height coordinates of the lidar was performed using the range weighting function (RWF) of the lidar in order to account for the spatial averaging inherent in the lidar measurements. This interpolation takes the following form:

$$\gamma_s(z) = \frac{\sum_{z_s} RWF(z_s - z)\gamma_s(z_s)}{\sum_{z_s} RWF(z_s - z)} \qquad (14)$$

where $\gamma_s(z_s)$ denotes either the $u$- or $v$-component of the sonic winds at the sonic height, $z_s$, and $\gamma_s(z)$ is the corresponding interpolated value at height z. The range weighting function is given by (Banakh and Smalikho 1997; Lundquist et al. 2015)

$$RWF(x) = \frac{1}{2\Delta r}\Big[ erf\left(f_+(x)\right) - erf\left(f_-(x)\right)\Big] \qquad (15)$$

where

$$f_\pm(x) = \frac{2\ln(2)}{\Delta p}\big| x / \sin\theta \pm \Delta r / 2\big|, \qquad (16)$$





$\Delta p$ is the laser pulse length, $\Delta r$ is the range gate length, and $\theta$ is the elevation angle. The laser pulse length for the lidar used in this study is $\Delta p \sim 22.5$m, and the range gate length was set to 30m. Figure 6 illustrates the range weightings for the four lidar range gates used in the comparison with the tower. It is clear that the range weighting has little effect on the sonic anemometer data because the instrument spacing is larger than the half-width of the weighting functions and the lidar range gate centers are close to the sonic heights. Thus, for this geometry, the application of equation (14) is essentially equivalent to using the nearest neighbor approximation.

Once the sonic data have been temporally averaged and vertically interpolated, the comparison with the lidar is carried out by computing statistics of the wind speed difference

$$\Delta_M = M - M_s, \tag{17}$$

and the wind direction difference,

$$\Delta_\alpha = \tan^{-1}\left(\frac{\sin\alpha\cos\alpha_s - \cos\alpha\sin\alpha_s}{\sin\alpha\sin\alpha_s + \cos\alpha\cos\alpha_s}\right), \tag{18}$$

where $\alpha = \tan^{-1}(u_o / v_o)$ is the azimuth angle of the horizontal velocity vector from the lidar, $M_s$ is the sonic anemometer wind speed, and $\alpha_s$ is the azimuth angle of the horizontal velocity vector as determined from the sonic anemometer data. The wind direction difference $\Delta_\alpha$ is positive when the lidar winds are rotated clockwise relative to the sonic winds. .We note that although equation (18) is mathematically equivalent to $\alpha - \alpha_s$, it is not prone to the problems that occur due to the cyclical nature of the angles.

Table 2 summarizes the results of the comparison between the CDL wind retrievals and the BAO tower for Trials 1, 2 and 3. These results represent averages taken over all four heights (142.9, 194.9, 246.8 and 298.8 m) and over the deployment period from 6 March through 16 April, 2015. Statistics were computed by excluding lidar measurements corresponding to wind speeds less than 0.5 ms$^{-1}$ in order to filter out less reliable wind direction data. The total sample count for each Trial was nominally 13000.

The wind speed bias is denoted by $\overline{\Delta_M}$ and the standard deviation of the wind speed difference is denoted by $\sigma(\Delta_M)$. Similarly, the wind direction bias is denoted by $\overline{\Delta_\alpha}$ and the standard deviation of the wind direction difference is denoted





by $\sigma(\Delta_\alpha)$. Also shown is the slope and offset (i.e. intercept) of the linear regression between the sonic anemometer and CDL-derived wind speeds, and the Pearson correlation coefficient, $r_{wspd}$, between the sonic and lidar wind speeds.

The results shown in Table 2 are divided into two data quality control categories. The first category uses no data rejection. In this category all of the measurements are used in the computation of the statistics, regardless of the estimated uncertainties.

The second category (i.e. last three columns of Table 2) shows the results with 50% data rejection. In this category, measurements with estimated relative wind speed uncertainties in the upper $50^{th}$ percentile are not used in the computation of the statistics.

Table 2 shows that wind speed biases range from -1 to 7 cm s$^{-1}$, and wind direction biases tend to cluster near -1°. We tested the significance of the wind speed biases from the difference distributions by computing the probability of obtaining a

measurement outside the range given by $\overline{\Delta_M} \pm \left|\overline{\Delta_M}\right|$. A similar test was conducted for the wind direction biases. In all cases, the probabilities were greater than 80%, with most probabilities above 90%. From this we conclude that the deviations from zero bias are not significant, so that the CDL-derived wind speed and direction measurements are essentially unbiased. Thus, it is appropriate to equate precision with uncertainty in this case.

Table 2 indicates that the wind speed and wind direction biases are insensitive to the treatment of the radial velocity

uncertainty in Equation (1) and are not affected significantly by data rejection. For 0% data rejection all three trials produce similar results. The differences between the trials are more evident when we compare results with and without data rejection. The results for trials 1 and 2 show significant improvement in the wind speed difference standard deviation, regression and correlation as the data rejection rate is increased from 0 to 50%. By contrast, Trial 3 shows no improvement in these quantities, suggesting that the uncertainty estimates for Trial 3 are poor indicators of data quality.

A common method of data quality control involves rejecting measurements whose estimated relative uncertainties exceed some prescribed threshold value. Figure 7a shows the mean percent difference in the wind speed , i.e. $100 \times \overline{\left|M - M_s\right| / M_s}$, as a function of the estimated relative uncertainty threshold. As this threshold is increased for a given trial, the mean percent wind speed difference asymptotically approaches a limiting value. Trial 3 converges to its limiting value much more quickly than trials 2 or 3. For a given threshold, trials 2 and 3 produced the smallest and largest

percent differences, respectively.

The application of the relative uncertainty threshold reduces the sample population, or data recovery. Fig 7b shows data recovery for Trials 1, 2 and 3 as a function of the relative uncertainty threshold. The data recovery is the percentage of measurements with estimated relative uncertainties below the prescribed threshold. For a given threshold level Trials 2 and 3 give the lowest and highest data recoveries, respectively. Trial 3 produces high data recoveries because the estimated

uncertainties are generally much lower than either Trial 1 or 2. Thus, for a given threshold level fewer measurements are rejected for Trial 3.





To assess the absolute accuracy of the uncertainty estimates, Fig 8a shows the relationship between the mean absolute wind speed difference and the estimated wind speed uncertainty, $\sigma_M$. Similarly, Fig 8b shows the mean absolute wind direction difference as a function of the estimated wind direction uncertainty, $\sigma_\alpha$. These curves were obtained by sorting the estimated uncertainties into discrete bins and computing the absolute wind speed and direction differences within each bin.

These results show that Trial 2 produced uncertainty estimates that agree well with the observed differences, and Trial 1 produces estimates that are about 30% smaller. By contrast, Trial 3 produces estimates that are far smaller than the observed differences, and compressed into a relatively narrow range of values.

## 4.2 Diurnal Variability

As seen in Figure 5, the estimated wind speed uncertainties for Trials 1 and 2 both show similar diurnal variations. To
investigate this dependence further, Fig 9a shows the mean absolute difference in wind speed and mean absolute wind direction difference between the CDL and the tower as functions of the time of day for Trial 2. These results were obtained by averaging over the deployment period from 6 March through 16 April, 2015. The lidar wind retrievals were quality controlled by rejecting measurements with estimated relative wind speed uncertainties greater than 25%. Also, Fig 9b shows the Turbulent Kinetic Energy (TKE) and the CDL data recovery as functions of the time of day. The TKE was computed
from the sonic anemometer data by taking the median value of 30-minute-averaged velocity variances over four levels on the tower (150, 200, 250, and 300m). We note that sunrise and sunset occurred at approximately 12 and 2 UTC, respectively.

Figure 9 shows that the mean absolute wind speed difference between the CDL and tower sonic anemometers is nominally around 30 cm s$^{-1}$ at night and then increases to about 90 cm s$^{-1}$ near solar noon. Similarly,  the mean absolute wind direction difference is roughly 5$^{\circ}$ at night and then increases to about 12$^{\circ}$ near solar noon. The diurnal variation in the wind speed and
wind direction differences is roughly correlated with the TKE (Fig 9b). The data recovery is approximately anti-correlated with the TKE. The data recovery reaches a maximum of just over 90% at night, and a minimum of between 60 to 70% during the day.

## 4.2 Effect of scan geometry and retrieval dimensionality

In this section we examine the effect of scan geometry and dimensionality (2D vs 3D) on the retrieved winds and their
uncertainties. Trials were conducted in which the retrieval algorithm was configured to perform either 2D or 3D retrievals using different subsets of the available pointing directions from each PPI scan. Table 3 provides a description of the configuration used for each of these Trials. All of these Trials use the same method for estimating uncertainty as Trial 2.

For the 2D retrievals the vertical velocity is assumed to be zero, so that only $u_o$ and $v_o$ are retrieved. Trials 2a, 2a$_{2D}$, 2b and 2b$_{2D}$ were performed to test the impact of altering the scan geometry by using subsets of the available pointing directions





from each PPI scan. For Trials 2a and $2a_{2D}$ we test the impact of eliminating pointing directions with complimentary azimuth angles (see Fig 2). Similarly, Trials 2b and $2b_{2D}$ use only orthogonal and anti-parallel pointing directions. Here, we refer to the scan geometry for Trials 2 and $2_{2D}$ as 8-beam full PPI scans. Trials 2a and $2a_{2D}$ are referred to as 4-beam sector PPI scans, and Trials 2b and $2b_{2D}$ are referred to as 4-beam full PPI scans.

Table 3 summarizes the results of the comparison between the lidar and the BAO tower for Trials 2, $2_{2D}$, 2a, $2a_{2D}$, 2b and $2b_{2D}$. As in Table 2, these results represent averages taken over all four heights (142.9, 194.9, 246.8 and 298.8 m) and over the deployment period from 6 March through 16 April, 2015. The lidar measurements were quality controlled using an estimated relative wind speed precision threshold of 25%.

Table 3 shows that the 2D retrievals generally resulted in slightly larger values of the wind speed and wind direction
difference standard deviations and slightly smaller values of the regression slopes and correlation coefficients when compared to their 3D counterparts. We also note that the 2D retrieval for the 4-beam sector PPI scan (Trial $2a_{2D}$) resulted in a much higher data recovery compared to the corresponding 3D retrieval (Trial 2a). Over all, the 2D 4-beam sector PPI scan (Trial $2a_{2D}$) resulted in the poorest agreement with the tower, and the 3D 8-beam full PPI scan (Trial 2) resulted in the best agreement with the tower.

**5 Summary**

This study evaluated the accuracy of wind speed and wind direction precision estimates as computed from CDL scan data using standard error propagation techniques in a VAD algorithm. Precision estimates were compared to differences in wind speed and direction between the VAD algorithm and the sonic anemometers mounted on the 300-m BAO tower. Three wind retrieval trials were conducted using different schemes for estimating the uncertainty in the retrieved wind speed and
direction. Comparisons were carried out using data from the XPIA field campaign in which one of the ARM CDLs was deployed in close proximity to the BAO tower.
For Trial 1 the radial velocity uncertainties were assumed to be unknown, so that the uncertainties in the retrieved velocity components were estimated from the diagonal elements of the unweighted covariance matrix and the fit residual, ie. equation (7). This approach is equivalent to assuming the radial velocity uncertainty to be isotropic. Trial 2 was a proxy for a PPI
sampling strategy in which multiple profiles are acquired for each pointing direction. This method allows for direct computation of the radial velocity variance, and avoids the assumption of isotropy inherent in Trial 1. For Trial 3 the effects of turbulence were completely ignored, and the radial velocity uncertainty was set equal to the radial velocity measurement precision. Since the measurement precision is parameterized in terms of the SNR, the radial velocity uncertainty for Trial 3 was determined solely by the SNR.
Our results showed that all trials produced qualitatively similar wind fields, with negligible bias when compared to the BAO tower sonic anemometers. There were, however, substantial differences in the precision estimates between the three trials.



The precisions for Trials 1 and 2 were well correlated but different in magnitude, with Trial 1 producing estimates that were about 30% smaller than Trial 2. Both Trials 1 and 2 exhibited a similar diurnal variation, with larger precisions occurring during the daytime under convective conditions. By contrast, the precision estimates for Trial 3 were much smaller in magnitude and poorly correlated with either Trials 1 or 2. We also note that since the results for all trials are unbiased, it is

appropriate to equate precision to uncertainty.

Trial 2 resulted in the best agreement between the estimated wind speed and direction precision estimates from the VAD algorithm and the mean absolute difference in wind speed and direction between the VAD algorithm and the tower. Filtering the retrieved winds based on the relative estimated wind speed precision resulted in a substantial improvement in the agreement between the retrieved winds and the tower for Trials 1 and 2, and little improvement for Trial 3. This behaviour

suggests that the precision estimates from Trial 3 are poor indicators of data quality.

An important lesson here is that it is inappropriate to equate the radial velocity uncertainty, $\sigma_r$, to the radial velocity precision $\sigma_n$ in the wind retrieval algorithm In the context of the VAD algorithm, turbulent fluctuations are a source of error, so ignoring this error results in biased wind speed and direction uncertainties that poorly represent the actual uncertainties. Best results are achieved when $\sigma_r$ can be determined directly from the standard deviation of the radial

velocity along each pointing direction and range gate of the PPI scan.

This study also investigated the effect of scan geometry and dimensionality (2D vs 3D) on the retrieved winds and their uncertainties. Trials were conducted in which the VAD algorithm was configured to perform either 2D or 3D retrievals using different 4-beam subsets of the available 8 beams from each PPI scan. We found that the 2D 4-beam sector PPI scan (Trial $2a_{2D}$) resulted in the poorest agreement with the tower, and the 3D 8-beam full PPI scan (Trial 2) resulted in the best

agreement with the tower.

The uncertainty estimation scheme of Trial 1 represents the current method that is used operationally by the ARM program for computing wind profiles from the all ARM CDLs (these data are publically available through the ARM web site at http://www.arm.gov/). This method is simple to implement, produces reasonable results, and requires only a standard "single-pass" PPI scan to run. Another important advantage is that the single-pass PPI scan can be executed quickly, thereby

minimizing interruptions to vertical velocity profiling, which is a high scientific priority among ARM data users.

Further testing is required in order to evaluate scanning strategies that enable direct computation of the radial velocity variance (as in Trial 2) while minimizing the impact on the vertical velocity profiling. Practical implementation could be achieved using a contiguous sequence of PPI scans with acquisition of a single profile in each pointing direction, or by acquiring multiple profiles along a given pointing direction before moving on to the next pointing direction. The latter

method is more efficient since it minimizes the downtime associated with moving the scanner from one position to the next, thereby minimizing the total scan time. Additionally, there are other trade-offs to be considered such as reducing the pulse





integration time, and perhaps reducing the number of pointing directions (while retaining at least 4 or more pointing directions), with the goal of keeping the total scan time to about 1 minute or less.

## Acknowledgements

This research was supported by the Office of Biological and Environmental Research of the U.S. Department of Energy as part of the Atmospheric Radiation Measurement Climate Research Facility. Funding for XPIA was provided by the U. S. Department of Energy, Office of Energy Efficiency and Renewable Energy, Wind and Water Power Technologies Office, and by NOAA's Earth System Research Laboratory. We express our appreciation to the National Science Foundation's CABL program (https://www.eol.ucar.edu/field_projects/cabl ) for supporting the deployment and operation of the sonic

anemometers on the BAO tower. We also thank Scott Sandberg, Aditya Choukulkar, and Paul Quelet for their assistance with the deployment of the ARM Doppler lidar.

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



**Table 1: Radial velocity and retrieved parameter precision estimation schemes for Trials 1, 2 and 3.**

| Trial | Radial Velocity Uncertainty | Retrieved parameter precision |
|---|---|---|
| 1 | $\sigma_{ri} = 1$ | Estimated from Equation (7) |
| 2 | $\sigma_{ri}$ is computed from 9 independent samples for each pointing direction | Estimated from Equation (6) |
| 3 | $\sigma_{ri} = \sigma_{ni}$ | Estimated from Equation (7) |





**Table 2: Results of the comparison between the lidar-derived winds and the sonic anemometers on the BAO tower. The results with 0% and 50% data rejection are shown. The averaging times for the sonic anemometer data was twice the PPI scan duration, or nominally about 80 seconds.**

| Data Rejection | 0% | | | 50% | | |
|---|---|---|---|---|---|---|
| Trial | 1 | 2 | 3 | 1 | 2 | 3 |
| $\overline{\Delta_M}$ (ms$^{-1}$) | 0.04 | 0.03 | 0.03 | -0.01 | 0.01 | 0.07 |
| $\sigma(\Delta_M)$ (ms$^{-1}$) | 0.7 | 0.69 | 0.7 | 0.4 | 0.55 | 0.75 |
| Regression offset (ms$^{-1}$) | 0.027 | 0.033 | 0.02 | -0.013 | -0.043 | -0.171 |
| Regression Slope | 0.987 | 0.988 | 0.991 | 1.004 | 1.005 | 1.015 |
| $r_{wspd}$ | 0.978 | 0.978 | 0.978 | 0.993 | 0.987 | 0.976 |
| $\Delta_\alpha$ (deg) | -0.78 | -0.73 | -0.86 | -1.17 | -1.07 | -0.51 |
| $\sigma(\Delta_\alpha)$ (deg) | 19.05 | 19.25 | 19.14 | 5.95 | 6.68 | 12 |





**Table 3: Results of the comparison between the CDL-derived winds and the tower for Trials 2, 2$_{2D}$, 2a$_{2D}$, 2b and 2b$_{2D}$. All Trials use the same method of precision estimation as Trial 2, and the results from each trial were quality controlled using an estimated relative wind speed precision threshold of 25%.**

| Description | 8-beam full PPI | | 4-beam sector PPI | | 4-beam full PPI | |
|---|---|---|---|---|---|---|
| Trial | 2 | 2$_{2D}$ | 2a | 2a$_{2D}$ | 2b | 2b$_{2D}$ |
| Dimensionality | 3D | 2D | 3D | 2D | 3D | 2D |
| Beams Used | 1 through 8 | | 1, 2, 3, 4 | | 2, 4, 6, 8 | |
| $\overline{\Delta_M}$ (ms$^{-1}$) | 0.06 | 0.05 | 0.16 | 0.24 | 0.09 | 0.1 |
| $\sigma(\Delta_M)$ (ms$^{-1}$) | 0.66 | 0.73 | 1.03 | 1.03 | 0.73 | 0.8 |
| Regression offset (ms$^{-1}$) | -0.046 | -0.03 | 0.221 | 0.08 | -0.037 | -0.031 |
| Regression Slope | 0.998 | 0.996 | 0.944 | 0.945 | 0.991 | 0.988 |
| $r_{wspd}$ | 0.98 | 0.976 | 0.957 | 0.954 | 0.976 | 0.971 |
| $\Delta_\alpha$ (deg) | -0.77 | -0.77 | 0.11 | -1.01 | -1.04 | -0.92 |
| $\sigma(\Delta_\alpha)$ (deg) | 10.48 | 12.14 | 11.45 | 13.16 | 9.49 | 10.98 |
| Data Recovery (%) | 85 | 86 | 44 | 74 | 71 | 73 |





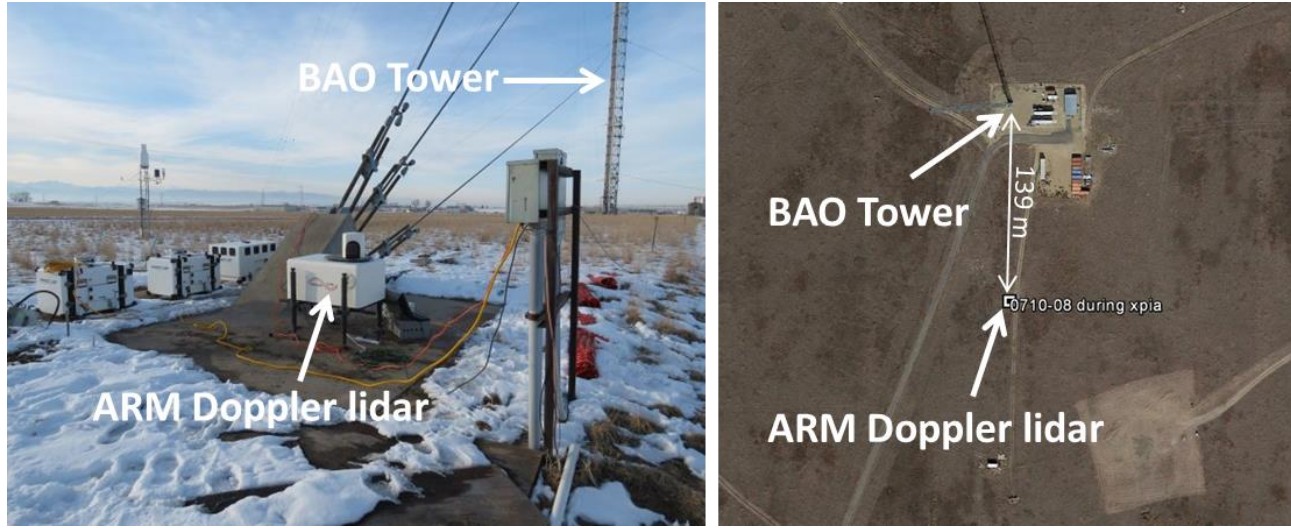

**Figure 1: a) Setup of the ARM Doppler lidar, and b) aerial view showing the location of the lidar relative to the BAO tower during XPIA.**



**Figure 2: Top view showing the azimuth angles of the eight-beam step-stare PPI scan pattern. Each beam is numbered for reference.**




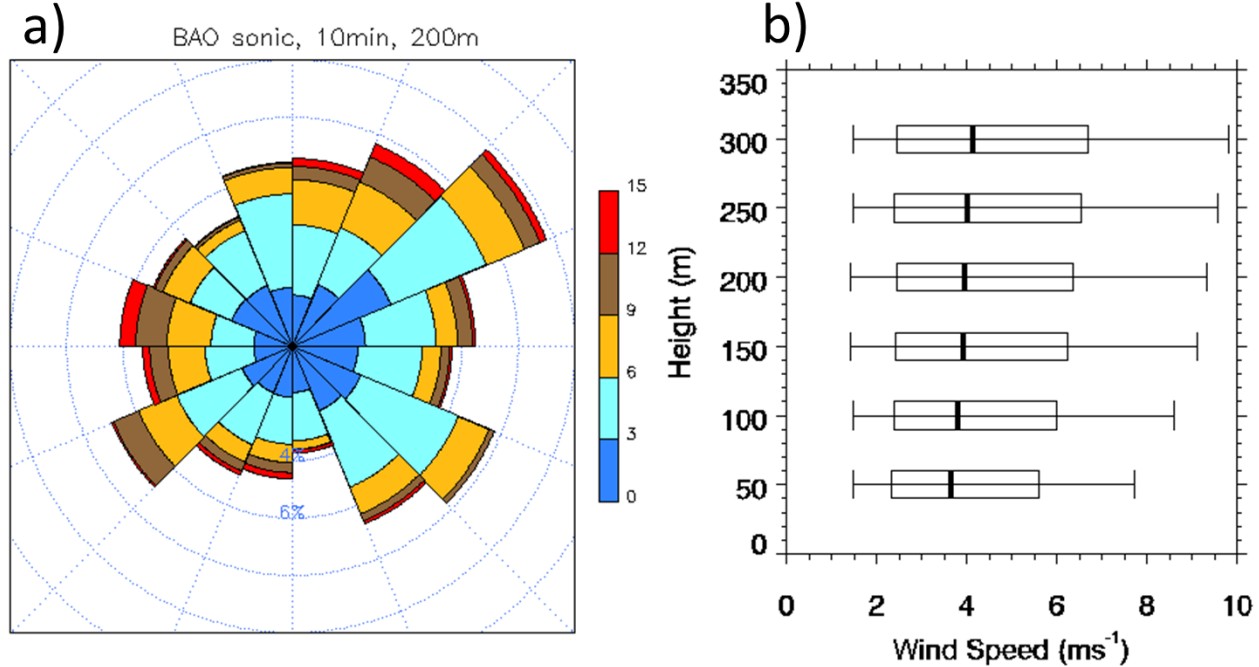

**Figure 3: . a) Wind rose from 10-min averaged sonic anemometer data at the 200m level on the BAO tower, and b) box and whisker plot of 10-min averaged wind speeds from the tower sonic anemometers. The box and whiskers indicate the 10th, 25th, 50th, 75th, and 90th percentiles. Both plots use data spanning the period from 6 March through 16 April, 2015.**



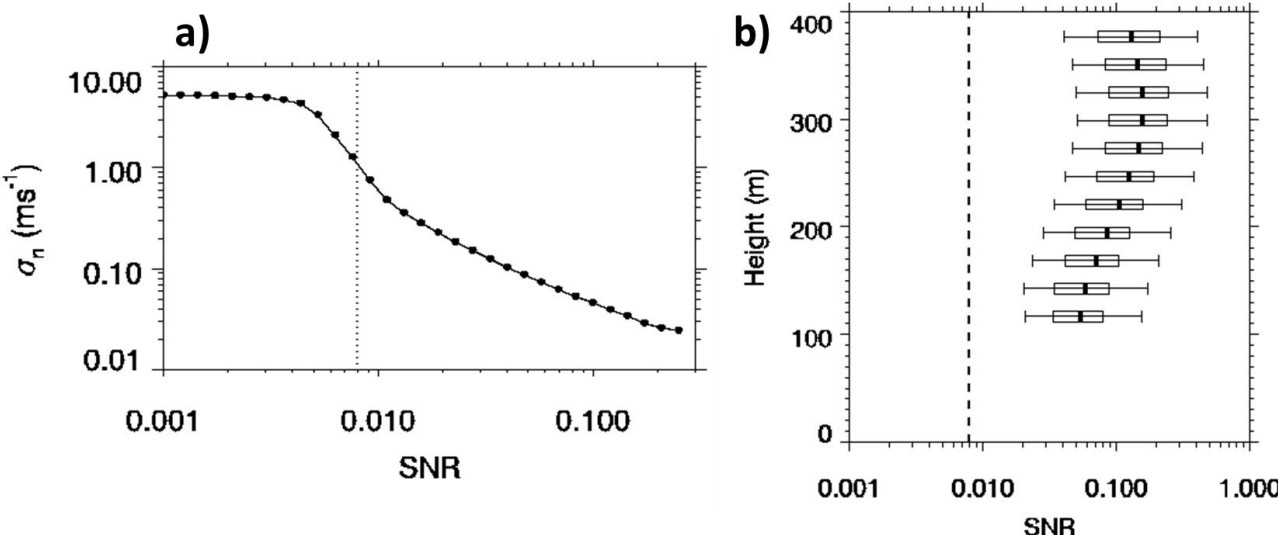

**Figure 4: a) The standard deviation of radial velocity measurement precision, $\sigma_n$, as a function of SNR, and b) box and whisker plot of the SNR profile indicating the 10th, 25th, 50th, 75th, 90th percentiles. Both plots use data spanning the period from 6 March through 16 April, 2015.**





**Figure 5: Time-height plots of the retrieved wind speed (left) and estimated wind speed precision (right) for 4 April 2015 for a) Trial 1, b) Trial 2, and c) Trial 3.**





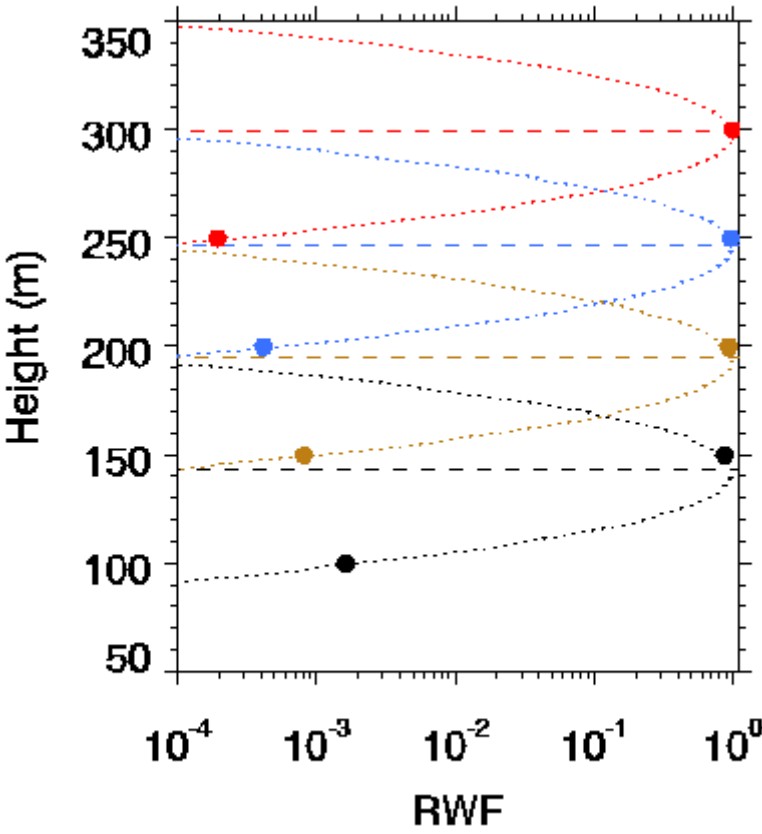

**Figure 6: Range weighting functions (RWFs) used to interpolate the sonic measurements to the heights of the lidar range gates. The dashed horizontal lines show the heights of the lidar range gates, and the solid filled circles show the heights of the sonic anemometers and the range-weighting values that were used in equation (14). Different colors are used to distinguish the lidar range gates and corresponding RWFs.**





**Figure 7: a) Mean percent difference in wind speed between the CDL and the tower, and b) the data recovery as functions of the relative precision threshold. The black, red and blue curves correspond to Trials 1, 2 and 3 respectively.**





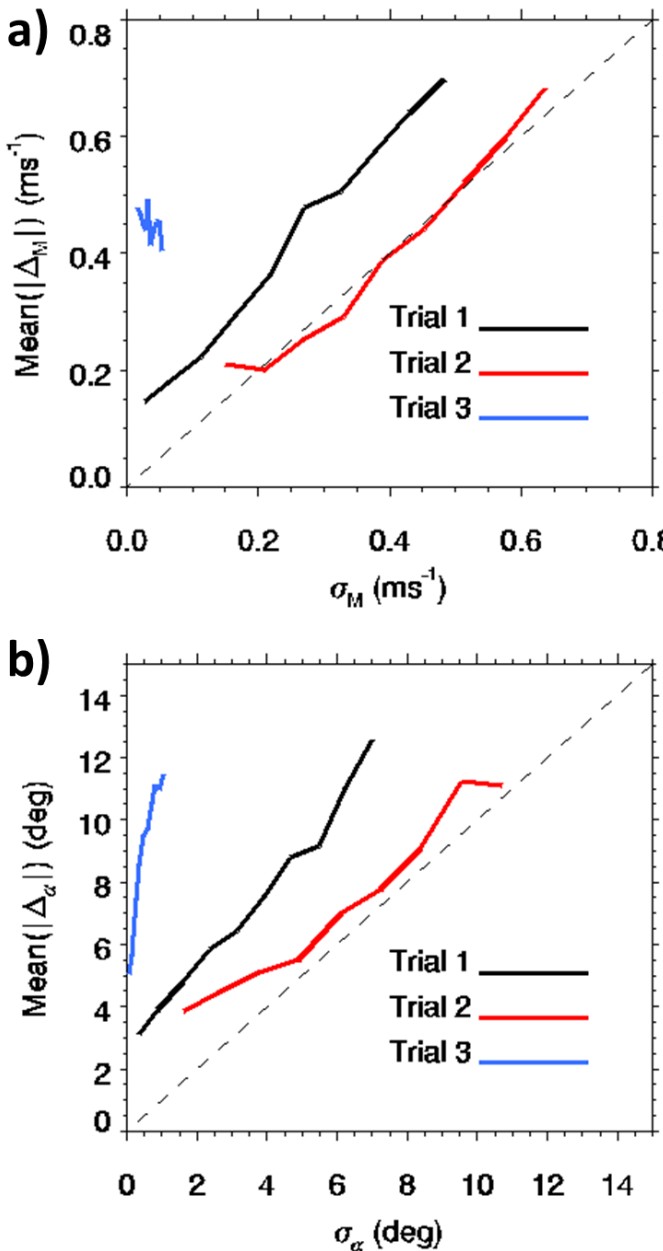

**Figure 8: a) Mean absolute difference in wind speed between the CDL and the BAO tower sonic anemometers as a function of the estimated wind speed precision, $\sigma_M$. b) Mean absolute difference in wind direction between the CDL and tower as a function of the estimated wind direction precision, $\sigma_\alpha$. The black, red and blue curves show the results for Trials 1, 2 and 3 respectively.**





**Figure 9: a) Diurnal variation of the mean absolute wind speed difference (blue) and the mean absolute wind direction difference (red) between the CDL and the tower sonic anemometers for Trial 2. b) Diurnal variation of the TKE (blue) and the data recovery (red). The CDL wind retrievals were quality controlled using a relative estimated precision threshold of 25%.**