# Peer review of "Validating Precision Estimates in Horizontal Wind Measurements from a Doppler Lidar"

_Atmospheric Measurement Techniques, 2016_

## Referee Comment (RC1) · Anonymous Referee #1 · 20 Dec 2016

Manuscript Number: AMT 2016-312
Title: Validating Precision Estimates in Horizontal Wind Measurements from a Doppler Lidar
Lead Author: Rob K. Newsom

**Summary**

In this paper, different methods are used to estimate precision in Doppler wind measurements from a scanning lidar deployed during XPIA. These precision estimates are compared to the actual horizontal wind speed and wind direction differences between the lidar and sonic anemometers on a tower to assess the ability of the various precision estimates to characterize lidar error. These types of precision estimates are extremely useful in the wind energy community, as they enable the calculation of uncertainty (if the lidar is unbiased) without measurements from a reference instrument.

Overall, the paper is clear and well-written. The clarification between uncertainty and precision is very helpful, although this clarification is sprinkled throughout the manuscript. It would be more helpful to state upfront (i.e., in the introduction section) what factors contribute to the random and systematic errors in Doppler lidar velocity data and how these terms relate to the definitions of uncertainty and precision.

The use of the terms uncertainty and precision is still a bit unclear in some parts of the manuscript. For example, in Section 3, $\sigma_{ri}$ is described as the measurement uncertainty due to random errors (p. 6, Line 2). But isn't uncertainty due to random errors just equivalent to precision? And in Section 4.1, you use the term uncertainty through p. 12, state that you can equate uncertainty with precision in this case because the CDL data are unbiased (p. 12, Lines 11-13), then continue to use the term uncertainty through the rest of Section 4.1. A clarification between these terms, perhaps with some additional symbols or equations, would be extremely useful. These terms are currently not very well-defined in the lidar literature, so laying out clear definitions of precision and uncertainty in the context of lidar measurements would make this paper a valuable reference.

Specific comments on the manuscript are listed below.

**Specific Comments**

Abstract

p. 1, Line 18: Should uncertainty be changed to precision here?
p.1, Lines 24-25: Briefly describe why ignoring turbulence effects results in uncertainty being equivalent to precision (i.e., how is turbulence defined in the context of random vs. systematic errors?)

**1. Introduction**

p. 3, Line 7: change "step-stair" to "step-stare"

**2. Experimental Setup and Instrumentation**

p. 5, Lines 12-18: I assume wind speeds from both sonics at each height were used to make these figures. How did you determine which sonic to use during each 10-minute period?

**3. Lidar Wind Retrieval and Precision Estimation**

p. 7, Line 5: What is $N$ in these equations?

p. 8, Lines 14-18: It looks like there's an SNR maximum at approximately 300 m. Does this correspond to the focus height of the lidar?

**4. Results**

p. 8, Lines 21-23: Make it clear from the beginning of the section what you are calculating the uncertainty of (10-min. wind speeds? Wind speeds from each 40-second VAD scan?)

p. 8, Line 25: Change "the all the" to "all the"

p. 9, Line 2: How far apart in time were the consecutive scans conducted? If they were spaced 10-15 minutes apart in time, how well do these variance measurements actually characterize atmospheric turbulence?

p. 9, Lines 15-16: This line (or a sentence with similar content) should be placed before the trials are described, so the reader is aware from the beginning that "trials" refer to different ways of processing the same data.

p. 9, Line 20: The use of the term "larger precisions" is misleading here. I would suggest changing it to something like "lower precision (higher uncertainty)".

p. 9, Lines 22-23: Briefly discuss why the uncertainty in Trial 3 shows no distinct diurnal variation.

p. 10, Lines 4-5: Briefly discuss why the averaging time for the sonics was set to twice the time of the PPI scans.

p. 10, Lines 5-6: It is a bit unclear what you mean by "under-sampling" here, and how this relates to the 12 min. PPI intervals and the 40-s scan time.

p. 10, Line 9: Change "rather the wind speeds…" to "rather than the wind speeds…"

p. 10, Lines 10-11: How did you define the wake sectors for the tower?

p. 11, Line 15: Delete extra period after "sonic winds"

p. 12, Lines 8-14: The significance of this statistical test is a bit unclear, so it would help to elaborate on the meaning of the test.

p. 12, Line 22: Please define the relative uncertainty (i.e., how is the uncertainty being normalized?) The term "relative uncertainty threshold" should be changed to "relative precision threshold" to be consistent with the terminology used in Fig. 7.

p. 13, Lines 17-22: Please discuss some possible reasons for the diurnal variability in wind speed and direction differences.

p. 14, Lines 5-7: Were there any noticeable differences in wind speed and direction correlations between the lidar and the tower for different heights or mean wind directions?

p. 14, Lines 9-14: Please elaborate on the significance of these findings and relate the different sub-trials to current lidar scanning techniques. For example, Trial $2a_{2D}$ is similar to low-elevation scans conducted by scanning lidars to measure, e.g., turbine wakes (although the scanning angle used for this CDL was quite high). Trial 2b is similar to a DBS scan used by a vertically profiling lidar. Practically speaking, at what point is the trade-off between decrease in scanning time and increase in uncertainty worthwhile? It should also be noted that if only 4 beams are used in the PPI scan rather than 8, the scan time would be much faster and it would be more feasible to conduct multiple contiguous PPI scans and/or spend more time collecting data at each azimuth angle.

It's a bit surprising that there isn't a significant change in the mean wind speed difference and wind speed difference standard deviation when the $w$ wind component is neglected. I would expect a noticeable change in uncertainty, as the $w$ component contributes significantly to the radial velocity at this PPI elevation angle.

5. Summary

p. 15, Line 12: Add period after "wind retrieval algorithm"

Tables and Figures

Table 1: It would help to add a column or two to this table to summarize the major assumptions made in each trial.

Figure 3a: Gridlines in the background should be made darker.

Figure 4: It would help to also give values of SNR in decibels on the x-axis of these plots, as SNR is often given in decibels in the lidar community.

---

## Referee Comment (RC2) · Anonymous Referee #2 · 23 Dec 2016

General comments:

Very interesting and well written paper on how the precision of VAD scans can best be quantified using in-situ data.

Following (maybe) my comments from the pre-review, the authors now make a clearer distinction between precision and uncertainty, where the latter contains contributions from both random ("precision", zero mean) and non-random ("bias", non-zero mean) sources. This paper, as the authors now recognise, is very much about precision and only mentions bias in passing. In their example, where the VAD scan is short (not the main duty of the lidar), this might well be fair enough since the random errors will be large and may (or may not) dominate. For a dedicated VAD scanner where the reconstructions are based on 10 minute mean radial wind speeds, the random error will be extremely small. In any case for an application such as a resource assessment in wind energy, it is bias (from the radial speed itself, from the elevation angle, from the range) that matters since the final result will inevitably be aggregated from many hundreds or thousands of samples. Put simply, random errors average to zero, biases don't! I would welcome some reflections on these issues in the paper (e.g. in the introduction or in a discussion) on where the precision quantification techniques are relevant (these are probably there already) and where they are less relevant.

I am still a little disturbed by the term 'radial wind speed uncertainty' meaning a spread of deviations from the speed expected by the VAD model but have trouble finding something better: 'radial wind speed non-conformity' perhaps? In any case I would be grateful if the authors could be even clearer when they introduce the term in explaining exactly what is meant.

The paper is a little long and sometimes I got a bit lost, especially in section 3. Maybe some more sub-section headings would be useful. Also consider shortening the paper. Could the work on using the precision assessment for looking at different scanning strategies (2d, 3d, 4beam, 8beam etc.) be moved to a separate paper (it gets a bit lost here anyway)?

Specific comments:

P2, line 23: Maybe make it clearer that by 'perfectly homogeneous flow' you also mean a flow without any turbulence. Could for example just add "of turbulence and" making ".. in the absence of turbulence and measurement error.."

P4, line 27: I think you mean "resolution" not "precision" (enough of those already…) as in "angular resolution of the scanner"

End of section 2.1: You briefly mention pointing uncertainty but you don't estimate it (i.e. elevation angle uncertainty, azimuth uncertainty) and you don't even mention range uncertainty. If you are not sensing at the right height (you aren't exactly) you are not sensing the right speed. Another significant uncertainty comes from the size and shape of the probe (you touch on this later). It would be a really useful addition to the paper to make an estimate of how much (non-random) uncertainty all these things (and the los speed uncertainty and anything else) combine to (e.g. using GUM). It is not zero – it never can be. It is probably quite significant.

P7, line 1: please explain why sigma_n should be 1.

P7, equation 7: please define psi

P8. After line 2: Here it could be good to have a sub-section heading "Obtaining the radial velocity measurement precision". Just an example.

P10, line 3: Why scalar averaging? Will it make much difference over such a short time anyway?

P10-11 – the section on interpolating the sonics using the lidar weighting function: This section strikes me as really over-complicated and unnecessary since as you conclude, the lidar senses pretty much at the sonic heights anyway (would have been a silly experimental design if it didn't). What would be more interesting here are some reflections on what you are comparing with what. What role does the uncertainty (precision + bias) of the sonic play (and how big are these)?

P12, line 13: "Thus it is appropriate to equate precision with uncertainty in this case." – Completely disagree with this statement. You have one observation (assuming the non-random effects to be persistent throughout the campaign) and are comparing against something that is itself uncertain. One zero error does not mean that the uncertainty is zero. This comes again in the conclusion (p15, line 4).

---

## Author Comment (AC1) · 17 Jan 2017

**Repsonses to Reviewer 1's Comments**

(the author's responses are given in **RED** text)

Manuscript Number: AMT 2016-312

Title: Validating Precision Estimates in Horizontal Wind Measurements from a Doppler

Lidar

Lead Author: Rob K. Newsom

Summary

In this paper, different methods are used to estimate precision in Doppler wind measurements from a scanning lidar deployed during XPIA. These precision estimates are compared to the actual horizontal wind speed and wind direction differences between the lidar and sonic anemometers on a tower to assess the ability of the various precision estimates to characterize lidar error. These types of precision estimates are extremely useful in the wind energy community, as they enable the calculation of uncertainty (if the lidar is unbiased) without measurements from a reference instrument.

Overall, the paper is clear and well-written. The clarification between uncertainty and precision is very helpful, although this clarification is sprinkled throughout the manuscript. It would be more helpful to state upfront (i.e., in the introduction section) what factors contribute to the random and systematic errors in Doppler lidar velocity data and how these terms relate to the definitions of uncertainty and precision.

The use of the terms uncertainty and precision is still a bit unclear in some parts of the manuscript. For example, in Section 3, sri is described as the measurement uncertainty due to random errors (p. 6, Line 2). But isn't uncertainty due to random errors just equivalent to precision? And in Section 4.1, you use the term uncertainty through p. 12, state that you can equate uncertainty with precision in this case because the CDL data are unbiased (p. 12, Lines 11-13), then continue to use the term uncertainty through the rest of Section 4.1. A clarification between these terms, perhaps with some additional symbols or equations, would be extremely useful. These terms are currently not very well-defined in the lidar literature, so laying out clear definitions of precision and uncertainty in the context of lidar measurements would make this paper a valuable reference.

In the original manuscript we used the terms like "random uncertainty" or "uncertainty due to random errors" interchangeably with "precision." In the revised manuscript we have attempted to clarify that random uncertainty = precision in the introduction. We have also replaced "uncertainty" with "precision" where appropriate throughout the entire revised manuscript.

Specific comments on the manuscript are listed below.

Specific Comments

Abstract

p. 1, Line 18: Should uncertainty be changed to precision here?

We have changed "uncertainty" to "precision" here and in many other places throughout the revised manuscript.

p.1, Lines 24-25: Briefly describe why ignoring turbulence effects results in uncertainty being equivalent to precision (i.e., how is turbulence defined in the context of random vs. systematic errors?)

The idea we're trying express is described by equation (11), i.e. the square of the total uncertainty is the sum of the real atmospheric variance and the instrumental noise variance. If we *assume* that the atmospheric variance is zero, then the radial velocity uncertainty is equal to the instrumental precision, by equation (11). What we're trying to say here is that if you make this assumption you get really bad estimates for the wind speed and direction uncertainties. Since this is the abstract we have to keep things as concise as possible. So we have modified the last sentence of the abstract to read

"By contrast, when instrumental measurement precision is assumed to be the only contribution the radial velocity uncertainty, the retrievals resulted in wind speed and direction precisions were biased far too low and poor indicators of data quality."

1. Introduction

p. 3, Line 7: change "step-stair" to "step-stare"

Done

2. Experimental Setup and Instrumentation

p. 5, Lines 12-18: I assume wind speeds from both sonics at each height were used to make these figures. How did you determine which sonic to use during each 10-minute period?

In the revised paper we have added the following text to p5 lines 6-8:

"For each 10-min averaging interval we used the sonic anemometer on the upwind side of the tower, as tower wake effects were observed to be quite significant (McCaffrey, et. al. 2016)."

3. Lidar Wind Retrieval and Precision Estimation

p. 7, Line 5: What is N in these equations?

N is the number of beams (LOSs) in the PPI scan that are used to retrieve the wind profile. This should have been mentioned in the text after equation 1 – sorry about that. We have since added a brief description after equation 1 in the revised manuscript.

p. 8, Lines 14-18: It looks like there's an SNR maximum at approximately 300 m. Does this correspond to the focus height of the lidar?

No not necessarily. In this case the focus was set at infinity.

4. Results

p. 8, Lines 21-23: Make it clear from the beginning of the section what you are calculating the uncertainty of (10-min. wind speeds? Wind speeds from each 40-second VAD scan?)

We have modified the first paragraph in section 4.1 to read "Three trials were conducted to evaluate different methods for estimating the radial velocity uncertainty and the resultant uncertainty in the derived wind components. For all of these trials, winds were computed from the 40-s PPI scans and the wind retrieval algorithm was configured to retrieve all three components of the wind field and use all eight beams of the PPI scans. These trials are described in Table 1."

p. 8, Line 25: Change "the all the" to "all the"

Done

p. 9, Line 2: How far apart in time were the consecutive scans conducted? If they were spaced 10-15 minutes apart in time, how well do these variance measurements actually characterize atmospheric turbulence?

As stated previously in section 2.1 "…the instrument was operated using a fixed scan schedule consisting of PPI scans once every 12 minutes." However, it doesn't hurt to remind the reader here, so we have added the following sentence to page 9 lines 4-5 in the revised manuscript:

"In the present case, the PPI scans are spaced 12 minutes apart, so we assume the atmosphere to be statistically stationary over a 24-minute period."

p. 9, Lines 15-16: This line (or a sentence with similar content) should be placed before the trials are described, so the reader is aware from the beginning that "trials" refer to different ways of processing the same data.

We have deleted the two sentences that appeared on p9 lines 15-16 of the original manuscript. Instead, we have added the following sentences to the end of the first paragraph of section 4.1 in the revised manuscript:

"All three trials use the same PPI scan data. The only difference between the trials is in the treatment of the radial velocity precision, $\sigma_r$ ."

p. 9, Line 20: The use of the term "larger precisions" is misleading here. I would suggest changing it to something like "lower precision (higher uncertainty)".

In this paper we have defined the precision to be the uncertainty due to random (zero-mean) error. So using this definition "high precision" means "high random uncertainty," which I admit sounds strange. To avoid this conundrum we have changed "larger precisions" to "larger random uncertainties."

p. 9, Lines 22-23: Briefly discuss why the uncertainty in Trial 3 shows no distinct diurnal variation.

We have added a sentence to the end of this paragraph (p9 lines 17-20 of the revised manuscript) to address the reviewers comment. The sentence reads:
 "The reason for this difference is that the radial velocity uncertainties for Trial 3 were determined solely from the SNR (see Fig 4a). The SNR responds to variations in aerosol properties (including size distribution and number density) and is less affected than the radial velocity by variations in atmospheric stability."

p. 10, Lines 4-5: Briefly discuss why the averaging time for the sonics was set to twice the time of the PPI scans.

During the course of our study we tested several averaging intervals ranging from 1x to 4x the PPI scan time. The results were not affected significantly. Although we were close to the tower (~140m), we weren't strictly collocated. Had we been collocated we would have set the averaging interval to the match the PPI scan time. Given that we were 140m from the tower we opted to go with an averaging interval that was 2x the PPI scan time.

p. 10, Lines 5-6: It is a bit unclear what you mean by "under-sampling" here, and how this relates to the 12 min. PPI intervals and the 40-s scan time.

The sonic data averaging interval is about 80 sec. The center times of the sonic averaging intervals are the same as the center times of the PPI scans, which are performed once every 12 min, or 720 sec. By "under-sampling" we mean that there is no overlap between the averaging intervals. If there was any overlap we would call it "oversampling." To make this point clear we have added the following text to p9 line 26 of the revised manuscript:

"(i.e. no overlap between averaging intervals)"

p. 10, Line 9: Change "rather the wind speeds…" to "rather than the wind speeds…"

Done

p. 10, Lines 10-11: How did you define the wake sectors for the tower?

For each level on the tower and for each averaging interval we used data from the anemometer that was on the upwind side of the tower. We did this by computing the mean wind direction from the two anemometers (using vector averaging). The "upwind" sector was then defined to be the sector that was

within +/-90 deg of the mean wind direction. The anemometer that was within that sector was used in our comparisons. To include this information in the revised manuscript we rearranged this paragraph to read (p10 lines 1-6):

"To avoid tower wake effects, only those anemometers on the upwind side of the tower were used in the comparison with the CDL measurements. For each tower level and averaging interval, the upwind side was defined by the azimuth sector that was within ±90° of the mean wind direction, as determined froma vector average of the two sonic anemometers. The temporally averaged (upwind) sonic anemometer data were then interpolated to the heights of the lidar range gates closest to the sonic anemometer heights (these heights were 142.9, 194.9, 246.8 and 298.8 m). The interpolation was handled by interpolating the horizontal vector components, rather the wind speeds and direction."

p. 11, Line 15: Delete extra period after "sonic winds"

Done.

p. 12, Lines 8-14: The significance of this statistical test is a bit unclear, so it would help to elaborate on the meaning of the test.

After taking a second look at our statistical test we've come to conclusion that it doesn't actually prove anything about the statistical significance of the difference in the wind speed means. Instead, we have applied the standard "student t-test." Those results indicate that the differences are significant. We have modified the text on p12 lines5-12 as follows:

"Table 2 shows that wind speed biases range from -1 to 7 cm s$^{-1}$, and wind direction biases tend to cluster near -1°. A student's t-test for paired data (Press et al. 1988) suggests that these biases, albeit small, are statistically significant. Table 2 also shows that the wind speed and wind direction biases are insensitive to the treatment of the radial velocity uncertainty in Equation (1) and are not affected significantly by data rejection. For 0% data rejection all three trials produce similar results. The differences between the trials are more evident when we compare results with and without data rejection. The results for trials 1 and 2 show significant improvement in the wind speed difference standard deviation, regression and correlation as the data rejection rate is increased from 0 to 50%. By contrast, Trial 3 shows no improvement in these quantities, suggesting that the uncertainty estimates for Trial 3 are poor indicators of data quality."

p. 12, Line 22: Please define the relative uncertainty (i.e., how is the uncertainty being normalized?) The term "relative uncertainty threshold" should be changed to "relative precision threshold" to be consistent with the terminology used in Fig. 7.

Done.

p. 13, Lines 17-22: Please discuss some possible reasons for the diurnal variability in wind speed and direction differences.

The wind speed and direction differences generally increase during the daytime due to the higher turbulence levels. In the original manuscript we point out that "The diurnal variation in the wind speed

and wind direction differences is roughly correlated with the TKE." To help clarify this point we have modified the text as follows (p13 lines 12-14 of the revised manuscript):

"The observed wind speed and wind direction differences increase as turbulence levels increase. Thus, the diurnal variation in the wind speed and wind direction differences are strongly correlated with the diurnal variation in TKE (Fig 9b)."

p. 14, Lines 5-7: Were there any noticeable differences in wind speed and direction correlations between the lidar and the tower for different heights or mean wind directions?

No, not over the relatively shallow layer (~150m to ~300m) that our analysis is constrained to.

p. 14, Lines 9-14: Please elaborate on the significance of these findings and relate the different sub-trials to current lidar scanning techniques. For example, Trial 2a2D is similar to low-elevation scans conducted by scanning lidars to measure, e.g., turbine wakes (although the scanning angle used for this CDL was quite high). Trial 2b is similar to a DBS scan used by a vertically profiling lidar. Practically speaking, at what point is the trade-off between decrease in scanning time and increase in uncertainty worthwhile? It should also be noted that if only 4 beams are used in the PPI scan rather than 8, the scan time would be much faster and it would be more feasible to conduct multiple contiguous PPI scans and/or spend more time collecting data at each azimuth angle. It's a bit surprising that there isn't a significant change in the mean wind speed difference and wind speed difference standard deviation when the w wind component is neglected. I would expect a noticeable change in uncertainty, as the w component contributes significantly to the radial velocity at this PPI elevation angle.

At the suggestion of the other reviewer we have opted to omit this material from our revised manuscript.

5. Summary

p. 15, Line 12: Add period after "wind retrieval algorithm"

Done

Tables and Figures

Table 1: It would help to add a column or two to this table to summarize the major assumptions made in each trial.

I believe the main differences/assumptions for each Trial are already summarized in the Table. The only differences are in the way that the radial velocity uncertainties are handled and the way the u and v uncertainties are computed.

Figure 3a: Gridlines in the background should be made darker.

Done.

Figure 4: It would help to also give values of SNR in decibels on the x-axis of these plots, as SNR is often given in decibels in the lidar community.

We have the changed the log scale to a dB scale on both plots in Fig 4.

---

## Author Comment (AC2) · 17 Jan 2017

**Responses to Reviewer 2's comments**

(author responses are given in **RED** text)

General comments:

Very interesting and well written paper on how the precision of VAD scans can best be quantified using in-situ data. Following (maybe) my comments from the pre-review, the authors now make a clearer distinction between precision and uncertainty, where the latter contains contributions from both random ("precision", zero mean) and non-random ("bias", non-zero mean) sources. This paper, as the authors now recognise, is very much about precision and only mentions bias in passing. In their example, where the VAD scan is short (not the main duty of the lidar), this might well be fair enough since the random errors will be large and may (or may not) dominate. For a dedicated VAD scanner where the reconstructions are based on 10 minute mean radial wind speeds, the random error will be extremely small. In any case for an application such as a resource assessment in wind energy, it is bias (from the radial speed itself, from the elevation angle, from the range) that matters since the final result will inevitably be aggregated from many hundreds or thousands of samples. Put simply, random errors average to zero, biases don't! I would welcome some reflections on these issues in the paper (e.g. in the introduction or in a discussion) on where the precision quantification techniques are relevant (these are probably there already) and where they are less relevant.

I am still a little disturbed by the term 'radial wind speed uncertainty' meaning a spread of deviations from the speed expected by the VAD model but have trouble finding something better: 'radial wind speed non-conformity' perhaps? In any case I would be grateful if the authors could be even clearer when they introduce the term in explaining exactly what is meant.

*In the introduction (original and revised) we felt it was important to stress that, in the context of VAD, any deviation in the radial velocity from a perfect sinusoid (when view as a function of the azimuth) is interpreted as error.*

The paper is a little long and sometimes I got a bit lost, especially in section 3. Maybe some more sub-section headings would be useful. Also consider shortening the paper. Could the work on using the precision assessment for looking at different scanning strategies (2d, 3d, 4beam, 8beam etc.) be moved to a separate paper (it gets a bit lost here anyway)?

*After some careful thought, we have to agree with the reviewer. We originally included the stuff on the effects of scan geometry and dimensionality in this paper because we felt it would be of interest. However, it doesn't fit too well. So, we have opted remove this material in the revised manuscript.*

Specific comments:

P2, line 23: Maybe make it clearer that by 'perfectly homogeneous flow' you also mean a flow without any turbulence. Could for example just add "of turbulence and" making ".. in the absence of turbulence and measurement error.."

*We have made the change.*

P4, line 27: I think you mean "resolution" not "precision" (enough of those already…) as in "angular resolution of the scanner"

Actually, it should be "precision" (i.e. random uncertainty in the pointing direction) and not resolution.

End of section 2.1: You briefly mention pointing uncertainty but you don't estimate it (i.e. elevation angle uncertainty, azimuth uncertainty) and you don't even mention range uncertainty. If you are not sensing at the right height (you aren't exactly) you are not sensing the right speed. Another significant uncertainty comes from the size and shape of the probe (you touch on this later). It would be a really useful addition to the paper to make an estimate of how much (non-random) uncertainty all these things (and the los speed uncertainty and anything else) combine to (e.g. using GUM). It is not zero – it never can be. It is probably quite significant.

As the reviewer points out, the issue of pointing accuracy is addressed in the last paragraph of section 2.1, where we discuss the daily target scans that were performed to monitor any azimuthal drift in the scanner. We did not observe any drift over the short deployment period (if we did we would have corrected for it). The daily target scans were also useful in establishing the so-called range errors. The plot below shows the SNR from a typical daily target scan. The "x" indicates the predicted location of the target (stadium light post) based on the known GPS coordinates of the target and the lidar. The red pixels show the high SNR returns from the light posts. As you can see, the "x" falls nearly in the middle (in the range dimension) of the pixel with the hard target return. Based on this we can safely assume the range error to be negligible.

[Figure]

In response to the reviewer's comment we have added the following text to p4 lines 24-26 of the revised manuscript:

"The observed location of the hard target return in the scan data, together with the known GPS coordinates of the lidar and the target enabled us to determine of the lidar's orientation with respect to true north, and to estimate any error in the reported range. In this case, no significant range errors were observed."

P7, line 1: please explain why sigma_n should be 1.

Sigma_n is the "instrumental precision" as we now refer to it in the revised manuscript. We do not set sigma_n=1, rather we set sigma_ri=1. In this case, all the measurements are equally weighted. This is common practice when the uncertainties are not known (see Numerical Recipies).

P7, equation 7: please define psi

Psi is given by equation (1).

P8. After line 2: Here it could be good to have a sub-section heading "Obtaining the radial velocity measurement precision". Just an example.

We have added a new subsection header on p7 line 17.

P10, line 3: Why scalar averaging? Will it make much difference over such a short time anyway?

Scalar averaging of the wind speed is the common practice in the wind energy community.

P10-11 – the section on interpolating the sonics using the lidar weighting function: This section strikes me as really over-complicated and unnecessary since as you conclude, the lidar senses pretty much at the sonic heights anyway (would have been a silly experimental design if it didn't). What would be more interesting here are some reflections on what you are comparing with what. What role does the uncertainty (precision + bias) of the sonic play (and how big are these)?

We should point out that the application of the range-weighting-function (RWF) serves a dual purpose. First, it provides a way of interpolating the sonic data to the height coordinates of the CDL. Second, it accounts for the spatial averaging effect that is inherent in the CDL measurements.
The reviewer raises a good point though. In the original manuscript, we simply introduce the RWF without explaining the need to reconcile point measurements (from the sonic) with spatial averages from the CDL. Thus, we have added the following text to p10 lines 7-13 in the revised manuscript:

"The sonic anemometer'a probe volume is considerably smaller than that of the CDL, where for all practical purposes, we may regard the sonic anemometer as a point measurement. By contrast, the CDL measurements represent a convolution (in the range dimension) of the instantaneous (i.e. point) radial velocity with the laser pulse range weighting function (RWF) and the range gate length (Frehlich and Cornman 2002). The size of the lidar's probe volume is defined by the width of the Gaussian laser pulse and the transverse extent of the beam, which is roughly 10cm.
In an effort to account for the spatial averaging that is inherent in the CDL measurements we applied an estimate of the CDL's RWF to the sonic anemometer data. This was also used as a means of interpolating the sonic anemometer measurements to the height coordinates of the CDL. This interpolation takes the following form:"

P12, line 13: "Thus it is appropriate to equate precision with uncertainty in this case." – Completely disagree with this statement. You have one observation (assuming the non-random effects to be persistent throughout the campaign) and are comparing against something that is itself uncertain. One zero error does not mean that the uncertainty is zero. This comes again in the conclusion (p15, line 4).

This paragraph has been completely revised. In the revised paragraph we no longer make the above statement. We are assuming the sonic data to be "truth." In the summary (original and revised) we are careful to state that the CDL winds showed "… negligible bias when compared to the BAO tower sonic anemometers." We believe that we have properly qualified our statement by saying "…when compared to the BAO tower sonic anemometers."

---

## Author Comment (AC3) · 17 Jan 2017

[revised manuscript text omitted]

25 GPS coordinates of the lidar and the target enabled us to determine of the lidar's orientation with respect to true north, and to estimate any error in the reported range. In this case, no significant range errors were observed. For this experiment the target that was used was a tall stadium light post located next to the football field at Erie High School, at a distance of about 800m west of the lidar location.

**2.2 BAO Tower Sonic Anemometers**

30 During XPIA the BAO tower was instrumented at six levels (50, 100, 150, 200, 250, and 300m) with fast response (20Hz) 3-D sonic anemometers (Campbell CSAT3). Each level had two sonic anemometers, one mounted on a southeast boom (at a

heading angle of 154º) and one mounted on a northwest boom (at a heading angle of 334º). This improved the odds of obtaining measurements that were unaffected by the tower wake. Data from the sonic anemometers were tilt-corrected (Wilczak et al. 2001), and rotated into geographical coordinates, with positive $u$ toward the east, positive $v$ toward the north, and $w$ pointing in the corrected vertical direction.

5    Figure 3 summarizes mean wind statistics as determined from 10-min average sonic anemometer data at the 200 m level on the BAO tower during the ARM CDL deployment period from 6 March through 16 April, 2015. For each 10-min averaging interval we used the sonic anemometer on the upwind side of the tower, as tower wake effects were observed to be quite significant (McCaffrey, et. al. 2016). The wind rose shown in Fig 3a indicates that the strongest winds tended to blow from the northeast and the west. Although there was no strongly preferred wind direction, there was a slightly higher occurrence

10   of northeasterly flow during the deployment period. Winds were generally fairly light with the bulk of the wind speeds occurring between 3 and 6 ms$^{-1}$. Fig 3b indicates that the median wind speeds tended to be at or slightly below 4 ms$^{-1}$. Inspection of the data shows that wind speeds exceeding 15ms$^{-1}$ occurred for brief periods on 16-17 March, 24-25 March, 12 April and 15-16 April. Overall, however, the winds rarely exceeded 10 ms$^{-1}$ during the deployment period.

**3 Lidar Wind Retrieval and Precision Estimation**

15   CDL estimates of horizontal winds are computed from PPI scan data using a velocity-azimuth-display (VAD) algorithm based on the classic technique described by Browning and Wexler (1968). Assuming the flow to be horizontally uniform and steady at a given range gate or height above the ground, the wind velocity components are retrieved by fitting a sinusoid to the radial velocity data; the amplitude, phase and offset of the sinusoid determine the wind speed, wind direction and vertical velocity, respectively. This is equivalent to minimizing

$$\psi^2 = \sum_{i=1}^{N} \frac{\left( \mathbf{u}_o \tilde{\mathbf{r}}_i^T - u_{ri} \right)^2}{\sigma_{ri}^2} \qquad (1)$$

with respect to the components of the mean velocity vector, $\mathbf{u}_o = (u_o, v_o, w_o)$. In equation (1), $N$ is the number of beams in the PPI scan, $u_{ri}$ is a radial velocity measurement, $\sigma_{ri}$ is the measurement uncertainty due to random errors (i.e. precision), and $\tilde{\mathbf{r}}_i$ is a unit vector from the lidar to the measurement point, i.e. the beam pointing direction, and is given by

$$\tilde{\mathbf{r}}_i = \left( \sin \phi_i \cos \theta, \cos \phi_i \cos \theta, \sin \theta \right) \qquad (2)$$

where $\phi_i$ is the azimuth angle as measured clockwise from true north, $\theta$ is the (constant) elevation angle as measured from the horizontal plane, and $\tilde{\mathbf{r}}_i^T$ is the transpose of $\tilde{\mathbf{r}}_i$. The summation in equation (1) is performed over all the pointing directions in the PPI scan. Minimizing equation (1) with respect to the components of $\mathbf{u}_o$ results in a system of three equations and three unknowns, the solution of which can be expressed as

$$\mathbf{u}_o = \mathbf{Cb},\tag{3}$$

where

$$\mathbf{C} = \left( \sum_{i=1}^{N} \frac{\tilde{\mathbf{r}}_i^T \tilde{\mathbf{r}}_i}{\sigma_{ri}^2} \right)^{-1}\tag{4}$$

is the covariance matrix, and

$$\mathbf{b} = \sum_{i=1}^{N} \frac{u_{ri}}{\sigma_{ri}^2} \tilde{\mathbf{r}}_i^T.\tag{5}$$

Equation 3 determines the wind velocity components at a fixed range gate. The height of the range gate above ground level is given by $z = r\sin\theta$, where $r$ is the distance from the lidar to the range gate center. Wind profiles are then constructed by applying equation (3) to all range gates.

15  When the individual radial velocity precisions, $\sigma_{ri}$, are known the precision of the retrieved velocity components can be obtained from the diagonal elements of the weighted covariance matrix (Press et al. 1988), i.e.

$$\sigma_u = \sqrt{C_{11}} \text{ and } \sigma_v = \sqrt{C_{22}}.\tag{6}$$

When the radial velocity precisions are not known, the precisions in $u$ and $v$ can be estimated by setting $\sigma_{ri} = 1$ in equation (1). The precisions of the retrieved velocity components are then determined in the following manner (Press et al. 1988):

$$\sigma_u = \psi\sqrt{\frac{C_{11}}{N - N_f}} \text{ and } \sigma_v = \psi\sqrt{\frac{C_{22}}{N - N_f}},\tag{7}$$

where $N_f$ is the number of retrieved parameters, i.e. $N_f = 3$ for 3D retrievals ($u_o$, $v_o$ and $w_o$) or $N_f = 2$ for 2D retrievals ($u_o$ and $v_o$ only).

Given the precisions for $u_o$ and $v_o$, the estimated precision in the retrieved wind speed is given by

[revised manuscript text omitted]

5    minutes apart, so we assume the atmosphere to be statistically stationary over a 24-minute period. We emphasize that the averaging scheme described above is only used to estimate the radial velocity precision. The radial velocity measurements used in Trial 2 are the same as those used in the other Trials.

For Trial 3 the radial velocity uncertainty, $\sigma_r$, is taken to be equal to the instrumental measurement precision, $\sigma_n$, so that the effects of turbulence are completely ignored. Since the instrumental precision is parameterized in terms of the SNR (see Fig 4a),

10   the radial velocity uncertainty for Trial 3 is determined solely by the SNR.

Figure 5 shows representative lidar wind retrievals for 4 April 2015, with results for Trials 1, 2 and 3 shown in panels (a), (b) and (c), respectively. The wind speed and direction retrievals (left) for all three trials look qualitatively similar, but the precision estimates (right) are substantially different, particularly between Trial 3 and either Trial 1 or 2. The precision for Trials 1 and 2 both show a strong diurnal dependence with larger random uncertainties during the daytime period (sunrise

15   and sunset are about 12 and 2 UTC, respectively). The precision estimates for Trial 2 are larger than Trial 1 and exhibit more smoothing as a result of the averaging described by equation (12). By contrast, the precision estimates for Trial 3 are much smaller than either Trials 1 or 2 and exhibit a completely different structure with no distinctive diurnal variation. The reason for this difference is that the radial velocity precisions for Trial 3 were determined solely from the SNR (see Fig 4a). The SNR responds to variations in aerosol properties (including size distribution and number density) and is less affected by

20   variations in atmospheric stability.

To enable comparison with the CDL-derived winds, 20Hz data from each sonic anemometer on the BAO tower were averaged in time and then interpolated to the heights of the lidar range gates closest to the sonic anemometer heights. The temporal averaging procedure used scalar averaging for the wind speed and vector averaging for the wind direction. The center times of the averaging intervals were made to coincide with the center times of the PPI scans, and the durations of the

25   averaging intervals were set equal to twice the PPI scan durations. We note that this method results in an under-sampling (i.e. no overlap between averaging intervals) of the sonic anemometer data, since the time between PPI scans is 12 min and the nominal scan duration is about 40 sec.

To avoid tower wake effects, only those anemometers on the upwind side of the tower were used in the comparison with the CDL measurements. For each tower level and averaging interval, the upwind side was defined by the azimuth sector that was within ±90º of the mean wind direction, as determined from a vector average of the two sonic anemometers. The temporally averaged (upwind) sonic anemometer data were then interpolated to the heights of the lidar range gates closest to the sonic anemometer heights (these heights were 142.9, 194.9, 246.8 and 298.8 m). The interpolation was handled by interpolating the horizontal vector components, rather the wind speeds and direction.

The CDL measurements represent a convolution (in the range dimension) of the instantaneous radial velocity with the laser pulse range weighting function (RWF) and the range gate length (Frehlich and Cornman 2002). The size of the lidar's probe volume is defined by the width of the Gaussian laser pulse and the transverse extent of the beam, which is roughly 10cm for the ARM CDL in the near-range. By contrast, the sonic anemometer may be regarded as a point measurement.
In an effort to account for the spatial averaging that is inherent in the CDL measurements we applied an estimate of the CDL's RWF to the sonic anemometer data. This was also used as a means of interpolating the sonic anemometer measurements to the height coordinates of the CDL. This interpolation takes the following form:

$$\gamma_s(z) = \frac{\sum\limits_{z_s} RWF(z_s - z)\gamma_s(z_s)}{\sum\limits_{z_s} RWF(z_s - z)} \qquad (14)$$

where $\gamma_s(z_s)$ denotes either the *u*- or *v*-component of the sonic winds at the sonic anemometer height, $z_s$, and $\gamma_s(z)$ is the corresponding interpolated value at height z. The range weighting function is given by (Banakh and Smalikho 1997; Lundquist et al. 2015)

$$RWF(x) = \frac{1}{2\Delta r}\left[erf\left(f_+(x)\right) - erf\left(f_-(x)\right)\right] \qquad (15)$$

where

$$f_\pm(x) = \frac{2\ln(2)}{\Delta p}\left|x/\sin\theta \pm \Delta r/2\right|, \qquad (16)$$

$\Delta p$ is the laser pulse length, $\Delta r$ is the range gate length, and $\theta$ is the elevation angle. The laser pulse length for the lidar used in this study is $\Delta p \sim 22.5$m, and the range gate length was set to 30m. Figure 6 illustrates the range weightings for the

four lidar range gates used in the comparison with the tower. It is clear that the range weighting has little effect on the sonic anemometer data because the instrument spacing is larger than the half-width of the weighting functions and the lidar range gate centers are close to the sonic heights. Thus, for this geometry, the application of equation (14) is essentially equivalent to using the nearest neighbor approximation.

5   Once the sonic data have been temporally averaged and vertically interpolated, the comparison with the lidar is carried out by computing statistics of the wind speed difference

$$\Delta_M = M - M_s,$$  (17)

and the wind direction difference,

$$\Delta_\alpha = \tan^{-1}\left(\frac{\sin\alpha\cos\alpha_s - \cos\alpha\sin\alpha_s}{\sin\alpha\sin\alpha_s + \cos\alpha\cos\alpha_s}\right),$$  (18)

10   where $\alpha = \tan^{-1}\left(u_o / v_o\right)$ is the azimuth angle of the horizontal velocity vector from the lidar, $M_s$ is the sonic anemometer wind speed, and $\alpha_s$ is the azimuth angle of the horizontal velocity vector as determined from the sonic anemometer data. The wind direction difference $\Delta_\alpha$ is positive when the lidar winds are rotated clockwise relative to the sonic winds. We note that although equation (18) is mathematically equivalent to $\alpha - \alpha_s$, it is not prone to the problems that occur due to the cyclical nature of the angles.

15   Table 2 summarizes the results of the comparison between the CDL wind retrievals and the BAO tower for Trials 1, 2 and 3. These results represent averages taken over all four heights (142.9, 194.9, 246.8 and 298.8 m) and over the deployment period from 6 March through 16 April, 2015. Statistics were computed by excluding lidar measurements corresponding to wind speeds less than 0.5 ms$^{-1}$ in order to filter out less reliable wind direction data. The total sample count for each Trial was nominally 13000.

20   The wind speed bias is denoted by $\overline{\Delta_M}$ and the standard deviation of the wind speed difference is denoted by $\sigma(\Delta_M)$. Similarly, the wind direction bias is denoted by $\overline{\Delta_\alpha}$ and the standard deviation of the wind direction difference is denoted by $\sigma(\Delta_\alpha)$. Also shown is the slope and offset (i.e. intercept) of the linear regression between the sonic anemometer and CDL-derived wind speeds, and the Pearson correlation coefficient, $r_{wspd}$, between the sonic and lidar wind speeds.

The results shown in Table 2 are divided into two data quality control categories. The first category uses no data rejection. In this category all of the measurements are used in the computation of the statistics, regardless of the estimated precision. The second category (i.e. last three columns of Table 2) shows the results with 50% data rejection. In this category, measurements with estimated relative wind speed precisions in the upper $50^{th}$ percentile are not used in the computation of the statistics.

Table 2 shows that wind speed biases range from -1 to 7 cm s$^{-1}$, and wind direction biases tend to cluster near -1°. A student's t-test for paired data (Press et al. 1988) suggests that these biases, albeit small, are statistically significant. Table 2 also shows 
[revised manuscript text omitted]